# Discovery of Anti-Coronavirus Cinnamoyl Triterpenoids Isolated from *Hippophae rhamnoides* during a Screening of Halophytes from the North Sea and Channel Coasts in Northern France

**DOI:** 10.3390/ijms242316617

**Published:** 2023-11-22

**Authors:** Malak Al Ibrahim, Zachee Louis Evariste Akissi, Lowiese Desmarets, Gabriel Lefèvre, Jennifer Samaillie, Imelda Raczkiewicz, Sevser Sahpaz, Jean Dubuisson, Sandrine Belouzard, Céline Rivière, Karin Séron

**Affiliations:** 1University of Lille, CNRS, Inserm, CHU Lille, Institut Pasteur de Lille, U1019—UMR9017—Center for Infection and Immunity of Lille (CIIL), F-59000 Lille, France; malak.alibrahim.etu@univ-lille.fr (M.A.I.); lowiese.desmarets@ibl.cnrs.fr (L.D.); imelda.raczkiewicz.etu@univ-lille.fr (I.R.); jean.dubuisson@ibl.cnrs.fr (J.D.); sandrine.belouzard@ibl.cnrs.fr (S.B.); 2BioEcoAgro, Joint Research Unit 1158, University of Lille, INRAE, University of. Liège, UPJV, YNCREA, University of Artois, University Littoral Côte d’Opale, ICV—Institut Charles Viollette, F-59650 Villeneuve d’Ascq, France; zachee.akissi@univ-lille.fr (Z.L.E.A.); gabriel.lefevre@univ-lille.fr (G.L.); jennifer.samaillie@univ-lille.fr (J.S.); sevser.sahpaz@univ-lille.fr (S.S.)

**Keywords:** SARS-CoV-2, HCoV-229E, antiviral agents, halophytes, *Hippophae rhamnoides*, triterpenoids

## Abstract

The limited availability of antiviral therapy for severe acute respiratory syndrome coronavirus 2 (SARS-CoV-2) has spurred the search for novel antiviral drugs. Here, we investigated the potential antiviral properties of plants adapted to high-salt environments collected in the north of France. Twenty-five crude methanolic extracts obtained from twenty-two plant species were evaluated for their cytotoxicity and antiviral effectiveness against coronaviruses HCoV-229E and SARS-CoV-2. Then, a bioguided fractionation approach was employed. The most active crude methanolic extracts were partitioned into three different sub-extracts. Notably, the dichloromethane sub-extract of the whole plant *Hippophae rhamnoides* L. demonstrated the highest antiviral activity against both viruses. Its chemical composition was evaluated by ultra-high performance liquid chromatography (UHPLC) coupled with mass spectrometry (MS) and then it was fractionated by centrifugal partition chromatography (CPC). Six cinnamoyl triterpenoid compounds were isolated from the three most active fractions by preparative high-performance liquid chromatography (HPLC) and identified by high resolution MS (HR-MS) and mono- and bi-dimensional nuclear magnetic resonance (NMR). Specifically, these compounds were identified as 2-*O*-*trans*-*p*-coumaroyl-maslinic acid, 3β-hydroxy-2α-*trans*-*p*-coumaryloxy-urs-12-en-28-oic acid, 3β-hydroxy-2α-*cis*-*p*-coumaryloxy-urs-12-en-28-oic acid, 3-*O*-*trans*-caffeoyl oleanolic acid, a mixture of 3-*O*-*trans*-caffeoyl oleanolic acid/3-*O*-*cis*-caffeoyl oleanolic acid (70/30), and 3-*O*-*trans*-*p*-coumaroyl oleanolic acid. Infection tests demonstrated a dose-dependent inhibition of these triterpenes against HCoV-229E and SARS-CoV-2. Notably, cinnamoyl oleanolic acids displayed activity against both SARS-CoV-2 and HCoV-229E. Our findings suggest that *Hippophae rhamnoides* could represent a source of potential antiviral agents against coronaviruses.

## 1. Introduction

Coronaviruses belonging to the sub-family *Orthocoronavirinae* of the *Coronaviridae* family are enveloped viruses with a positive-strand RNA genome. To date, seven human coronaviruses (HCoVs) have been identified, including HCoV-NL63, HCoV-229E, HCoV-HKU1, HCoV-OC43, SARS-CoV, Middle East respiratory syndrome coronavirus (MERS-CoV), and SARS-CoV-2 [1]. Coronaviruses have shown high transmissibility potential with varying mortality rates. Four viruses, HCoV-NL63, -229E, -HKU1, and -OC43, are known to cause mild to moderate respiratory diseases in immunocompetent individuals [2]. The remaining three, SARS-CoV, MERS-CoV, and SARS-CoV-2, are highly pathogenic, with significantly higher mortality rates. To date, three coronavirus outbreaks have occurred, including the 2002–2003 SARS epidemic caused by SARS-CoV, the 2012 MERS epidemic caused by MERS-CoV, and the 2019-current SARS-CoV-2 pandemic (COVID-19) [3]. This pandemic has posed profound unprecedented challenges on the global economy and healthcare sectors. While several vaccines have been manufactured and licensed, nearly 4 years into the pandemic, only three drugs, paxlovid (nirmatrelvir and ritonavir), remdesivir, and molnupiravir, have been granted authorization by the Food and Drug Administration (FDA) [4]. However, the European Medicines Agency considered that the benefit/risk balance of molnupiravir in the treatment of COVID-19 had not been established and refused its marketing authorization [5]. Moreover, inequitable access to COVID-19 therapies and vaccines, particularly in lower-income countries, persists [6]. Thus, the need to search for new low-cost and effective antivirals against COVID-19 is crucial.

Some of the current studies on drug therapies against COVID-19 are investigating plant secondary metabolites for their potential antiviral capacity against SARS-CoV-2. According to the WHO, around 80% of the world’s population relies on traditional plants or herbs to fulfill their basic health needs. Several countries worldwide have started to study the antiviral role of their traditional medicinal plants in combating COVID-19 [6,7,8,9]. Secondary metabolites found in plants play a significant biological and ecological function, particularly in chemical defense because of their anti-oxidative and antimicrobial activities. Plants subjected to abiotic stresses such as soil drought and salinity are considered rich sources of bioactive molecules due to their anti-oxidative systems and biochemical and molecular mechanisms that can survive in abiotic environments. The separation of halophytes from more sensitive plants, called glycophytes, is most often based on the salt (NaCl) concentration, which was typically set at 86 mM (0.5% NaCl). However, more recent studies have suggested a limit of 200 mM NaCl in order to separate true halophytes (euhalophytes) from relatively tolerant species. Some species, such as *Tecticornia* spp., are capable of withstanding NaCl concentrations of 10 mM to 2 M [10,11,12]. A plethora of definitions have been attributed to halophytes due to their taxonomical and ecological complexity, and the definition remains debated. Succulence, a common feature in halophytes, is associated with the existence of salt secretory glands, which are modified trichomes that were originally epidermal cells. The coastline of northern France (Hauts-de-France region) harbors some halophilic vegetations that belong to different botanical families, including Amaranthaceae, Apiaceae, Asteraceae, Brassicaceae, Caryophyllaceae, Convolvulaceae, Cyperaceae, Euphorbiaceae, Gentianaceae, Juncaceae, Juncaginaceae, Papaveraceae, Plantaginaceae, Plumbaginaceae, Poaceae, Polygonaceae, Primulaceae, and Ruppiaceae [13]. Studies have shown that halophilic plants possess a broad spectrum of antiviral activity against different viruses, including herpes simplex virus-2, influenza virus, and adenoviruses, due to their specialized metabolites such as saponins, alkaloids, tannins, and flavonoids [13,14,15,16]. Moreover, we previously published that a phenanthrene derivative, dehydrojuncusol, isolated from a halophyte, *Juncus maritimus* L. (Juncaceae), has been shown to be an inhibitor of hepatitis C virus replication [17]. Hence, in our work, different halophytes and less salt-tolerant plants collected from northern France were first screened for their antiviral activity against human coronavirus HCoV-229E in vitro. The most active extracts and fractions were then tested against SARS-CoV-2 in order to identify potential pan-coronavirus antiviral agents. Bioguided fractionation was performed on the most active plant species, *Hippophae rhamnoides* L. (Eleagnaceae), in order to identify natural bioactive compounds.

## 2. Results

### 2.1. Sampling and Classification of the Collected Plant Species

Twenty-two plant species, including strictly halophytes and relatively salt-tolerant species, were selected and collected from five different locations (Étaples, Dannes, Le Portel, Gravelines, Zuydcoote) distributed across the coastline of the North Sea and the English Channel in northern France (Hauts-de-France region). The whole plant or, in some cases, different parts of the plants (leaves (L), stems (S), roots (R)) were powdered to produce twenty-five crude methanolic extracts. The majority of these plants are representative of the botanical families of salt-tolerant plants distributed on the coasts of the North Sea and the English Channel. Some of them are considered strictly halophytes and were found in a schorre or at the base of an incipient dune, such as the Amaranthaceae species, *Cakile maritima* Scop. subsp. *integrifolia* (Brassicaceae), and *Lysimachia maritima* (Primulaceae). The majority of the collected plants belong to the families of Asteraceae and Amaranthaceae, each representing 18% (*n* = 4) of all the plants collected. This repartition is logical with regard to Amaranthaceae since they are the most representative family of halophytes from the coast in the region [13]. Cyperaceae and Poaceae each represent 9% (*n* = 2). The remaining families (Apiaceae, Berberidaceae, Convolvulaceae, Brassicaceae, Elaeagnaceae, Euphorbiaceae, Onagraceae, Primulaceae, Rosaceae, and Salicaceae) each represent 4.5% (*n* = 1) of the collected plants (Figure 1 and Appendix A and Table 1).

### 2.2. Cytotoxicity and Antiviral Activity of Plant Crude Methanolic Extracts

#### 2.2.1. Effect of Crude Methanolic Extracts on Cell Viability

The cytotoxicity of 25 crude methanolic extracts was tested on Huh-7 cells using an [3-(4,5-dimethylthiazol-2-yl)-5-(3-carboxymethoxyphenyl)-2-(4-sulfophenyl)-2H-tetrazolium]- based (MTS) viability assay. Huh-7 cells were treated with two different concentrations of the crude methanolic extracts (25 and 100 μg/mL) for 24 h. Non-treated control cells were incubated with 0.1% dimethyl sulfoxide (DMSO) in the media. The concentration of 25 µg/mL of all crude methanolic extracts was tolerated by Huh-7 cells after 24 h treatment. Similarly, no cellular cytotoxicity was observed using a concentration of 100 µg/mL of the crude methanolic extracts on Huh-7 cells, except for *Convolvulus soldanella*, *Berberis aquifolium* (R), and *Lysimachia maritima*, causing a decrease in cell viability by 53%, 60%, and 72.6%, respectively (Figure 2).

#### 2.2.2. Antiviral Screening of the Plant Crude Methanolic Extracts on HCoV-229E

Following the identification of the toxicity, we studied the antiviral activity of the extract on HCoV-229E infection. A coronavirus enters the cells through one of two pathways: by endocytosis or by direct fusion with the plasma membrane. The host-cell protease transmembrane serine protease 2 (TMPRSS2) is necessary for the plasma membrane fusion of many coronaviruses, including HCoV-229E, whereas cathepsins are often involved in fusion processes at endosomal membranes [18]. We screened the antiviral activity of the crude methanolic extracts at a concentration of 25 µg/mL by quantifying the infection of HCoV-229E-Luc in Huh-7 cells, whether they expressed the TMPRSS2 protease or not. A significant decrease in the luciferase activity representing an antiviral effect was observed using *Hippophae rhamnoides*, *Salix repens* (R), *Salix repens* (S), *Berberis aquifolium* (R), and *Baccharis halimifolia* (L) in Huh-7/TMPRSS2 cells. Similarly, an antiviral effect was observed with *Hippophae rhamnoides*, *Salix repens* (R), *Salix repens* (S), *Berberis aquifolium* (R), and *Baccharis halimifolia* (L) in Huh-7 cells (Figure 3). Since *Berberis aquifolium* (R) crude methanolic extracts showed cytotoxicity at 100 µg/mL, we decided not to study it further.

### 2.3. Dose-Response Antiviral Activity of Plant Extracts

In order to confirm the antiviral activity of *Hippophae rhamnoides*, *Salix repens* (R), *Salix repens* (S), and *Baccharis halimifolia* (L) extracts on HCoV-229E, dose-response experiments were conducted. Antiviral assays were conducted in Huh-7 and Huh-7/TMPRSS2 cells to cover the two entry pathways. Cytotoxicity was also evaluated in parallel. The results presented in (Appendix A) show that the four selected extracts were able to decrease HCoV-229E infection in a dose-dependent manner, confirming their antiviral capacity. These results allowed us to determine the 50% cytotoxic concentration (CC_50_) and the 50% inhibitory concentration (IC_50_) of each extract and calculate their selectivity index (SI), which is the ratio between CC_50_ and IC_50_ (Table 2).

*Hippophae rhamnoides* showed the highest CC_50_ in Huh-7 cells (621 μg/mL), whereas *Salix repens* (R) showed the lowest CC_50_ of 149 μg/mL. For antiviral activity, *Salix repens* (S) demonstrated the lowest IC_50_ in both Huh-7 and Huh-7/TMPRSS2 cells (15.1 and 14.7 µg/mL, respectively). However, *Baccharis halimifolia* (L) extract was the most active among the extracts but only in Huh-7/TMPRSS2 cells, with an IC_50_ of 11.2 µg/mL, showing that it might inhibit the TMPRSS2 entry pathway. Finally, for the four extracts, antiviral activity was not due to cytotoxicity, with the calculated SIs ranging from 5 to 35.

As a next step, we wondered if the extracts could have antiviral activity against other HCoVs and examined their antiviral capacity against SARS-CoV-2 in Vero-81 cells. Cytotoxicity was first evaluated (Table 2, Appendix A). Vero-81 cells showed higher tolerance to the crude methanolic extracts compared to Huh-7 cells due to the presence of a P-glycoprotein efflux pump [19]. Vero-81 cells infected with the SARS-CoV-2 were treated with two concentrations, 25 and 50 μg/mL, of the crude methanolic extracts, and chloroquine, an inhibitor of the endocytic entry pathway, was added as control. Western blot analyses showed a dose-dependent decrease in the SARS-CoV-2 N protein expression levels for the four extracts, indicating an antiviral effect (Figure 4). Three crude methanolic extracts, *Hippophae rhamnoides*, *Salix repens* (R), and *Baccharis halimifolia* (L), showed a high antiviral effect at both concentrations. Even though *Salix repens* (S) showed an antiviral effect at 50 μg/mL, the antiviral activity at 25 μg/mL was weaker (Figure 4).

Taken together, these results show that the four crude methanolic extracts of salt-tolerant species might be a source of antiviral compounds against human coronaviruses HCoV-229E and SARS-CoV-2.

### 2.4. Bioguided Fractionation Assay to Determine the Active Sub-Extracts

A bioguided fractionation assay was conducted in order to isolate the active compounds present in the active crude methanolic extracts. The four extracts were partitioned using three solvents of different polarities, yielding dichloromethane (DCM), ethyl acetate (EtOAc), and aqueous (Aq) sub-extracts. The different partitions obtained were tested for cytotoxicity at 25 and 100 μg/mL through the MTS assay. None of the tested sub-extracts appeared to be cytotoxic at 25 μg/mL (Appendix A).

Then, the antiviral activity of each sub-extract at 25 µg/mL was tested against HCoV-229E (Figure 5). The DCM sub-extract was the most active for *Hippophae rhamnoides* in both Huh-7 and Huh-7/TMPRSS2 cells. The Aq sub-extract was the most active in *Salix repens* (S) in both Huh-7 and Huh-7/TMPRSS2 cells. All sub-extracts of *Salix repens* (R) showed an antiviral effect against HCoV-229E. Surprisingly, for *Baccharis halimifolia* (L), fractionation did not permit the identification of a very active sub-extract. We selected *H. rhamnoides* DCM, *S. repens* (R) EtOAc, *S. repens* (S) Aq, and *B. halimifolia* DCM for further investigations.

In order to confirm the antiviral activity of the selected sub-extracts and determine their cytotoxicity, dose-response experiments were performed as described. This allowed us to calculate the CC_50_, IC_50,_ and SI values (Table 3, Appendix A). All the tested sub-extracts showed a dose-dependent reduction in infection with a high SI (SI > 10). The EtOAc sub-extract of *Salix repens* (R) displayed the lowest IC_50_ in Huh-7/TMPRSS2 cells (IC_50_ = 28.8 µg/mL), whereas the Aq sub-extract of *Salix repens* (S) showed the lowest IC_50_ value in Huh-7 cells (IC_50_ = 7.6 µg/mL). On the other hand, the DCM sub-extract of *Baccharis halimifolia* showed the highest IC_50_ values of 38.3 µg/mL and 31.1 µg/mL in Huh-7/TMPRSS2 and Huh-7 cells, respectively.

We then tested the antiviral activity of the sub-extracts on SARS-CoV-2. A dose-dependent inhibitory effect was observed for all extracts (Figure 6). Treatment with the DCM sub-extract of *Hippophae rhamnoides* (HR DCM SE) at a concentration of 50 µg/mL led to a strong reduction in SARS-CoV-2 infection by showing a complete decrease in N protein expression levels in the cell lysate. The DCM sub-extract of *Baccharis halimifolia* showed the lowest inhibitory activity against SARS-CoV-2 at both concentrations. Both the EtOAc sub-extract of *Salix repens* (R) and the Aq sub-extract of *Salix repens* (S) showed similar inhibitory effects against SARS-CoV-2; however, it was lower than that of HR DCM SE.

Since the HR DCM SE showed a strong antiviral effect against both HCoV-229E and SARS-CoV-2, we decided to study this plant in depth to determine the major bioactive compounds responsible for the anti-coronavirus effect.

### 2.5. Characterization of HR DCM SE by UHPLC UV-MS

The analysis of HR DCM SE by UHPLC-UV-MS showed different compounds with different levels and polarities. HR DCM SE was fractionated by CPC with the Arizona R system in descending mode. The collected fractions were dried and analyzed by UHPLC-UV-MS. The fractions that showed the same chromatographic profile were pooled together. Thus, we ended up with 10 fractions (Appendix A).

### 2.6. Cytotoxicity of Ten Fractions Prepared from HR DCM SE

The cytotoxicity of the ten fractions of HR DCM SE obtained through CPC was assessed in Huh-7 cells and Vero-81 cells at 25 and 100 µg/mL. Generally, a dose-dependent decrease in cell viability was observed upon testing the fractions at increasing concentrations. However, at 25 µg/mL, all fractions showed no cytotoxicity on Huh-7 cells. At 100 µg/mL, fractions F2, F3, F4, F6, and F7 were considered cytotoxic upon treating the cells, causing a decrease in cell viability by 41.7%, 44.2%, 61.4%, 30.9%, and 70.4%, respectively. However, fractions F1, F2, F9, and F10, did not show cellular toxicity. Similar to Huh-7, these fractions were not cytotoxic on Vero-81 cells when tested at 25 µg/mL. Fractions F2, F4, F6, and F7, which showed cytotoxicity on Huh-7 cells, were also toxic on Vero-81 cells at 100 µg/mL, decreasing the viability by 55.7%, 75.7%, 81.7%, and 66.4%, respectively (Appendix A).

### 2.7. Antiviral Screening of Ten Fractions Obtained from HR DCM SE against HCoV-229E and SARS-CoV-2

After identifying a concentration that could be tolerated by Huh-7 cells, the antiviral activity on HCoV-229E-Luc in Huh-7 and Huh-7/TMPRSS2 cells was monitored. Huh-7 cells were infected with the virus and treated simultaneously with the different fractions at 10 and 25 μg/mL. Three fractions (F3, F4, and F7) showed a 10-fold decrease in virus infection levels compared to the control, indicating an antiviral effect against HCoV-229E (Figure 7A).

Similarly, the ten fractions were tested for antiviral activity against SARS-CoV-2. The results presented in (Figure 7B) show that, interestingly, fractions F2 and F6, which were not active on HCoV-229E, were able to inhibit the expression of the SARS-CoV-2 N protein. Similar to HCoV-229E, fractions F3, F4, and F7 were able to inhibit SARS-CoV-2 replication, showing a dose-dependent decrease in antiviral activity (Figure 7B).

### 2.8. Isolation of Compounds by Preparative HPLC and Identification by HRMS and NMR

To isolate the active compounds responsible for an antiviral effect against both viruses, a preparative HPLC was performed. Fractions F2, F4, and F7 were first analyzed by analytical HPLC to find the best purification method. We chose not to purify compounds from F3 due to their high complexity and because the compounds of interest were also found in the less complex fractions, F2 and F4 (Appendix A). Then, the major compounds were purified by preparative HPLC using the same stationary phase. Fraction 2 was purified with a gradient system (60–100% CH_3_CN, 30 min), providing compounds F2-1 (tr = 23.00 min, 2.41 mg), F2-2 (tr = 23.55 min, 2 mg), F2-3 (tr = 26.00 min, 2 mg), and F2-4 (tr = 9.830 min, 3 mg) (Figure 8). F2-1, F2-2, and F2-3 have a molecular mass of 618 g.mol^−1^. Two compounds, F4-1 (tr = 16.507 min, 3 mg) and F4-2 (tr = 18.946 min, 8 mg), with a molecular mass of 618 g.mol^−1^, were purified from F4 with a gradient system (80–100% CH_3_CN, 30 min) (Figure 8). Compound F4-1 was isolated as a mixture (30:70) with compound F4-2. F7 was fractionated with a gradient system (80–100% CH_3_CN, 30 min) to give compound F7-1 (tr = 19.030 min, 3.4 mg) with a molecular mass of 602 g.mol^−1^ (Figure 8).

Taken together, F2-1, F2-2, F3-2, F4-1, F4-2, and F7-1 were identified as six cinnamoyl triterpenoids. Their purities were, respectively, estimated at 98.8, 88.6, 97.1, 90.7, 96.4, and 91.7% on the basis of PDA chromatograms (Appendix A). Their structures were established through a comparison of their physical and spectral data, including HRMS and extensive 1D- and 2D-NMR data, with reported values of 2-*O*-*trans*-*p*-coumaroyl-maslinic acid (F2-1) [20], 3β-hydroxy-2α-*trans*-*p*-coumaryloxy-urs-12-en-28-oic acid (F2-2) [21], 3β-hydroxy-2α-*cis*-*p*-coumaryloxy-urs-12-en-28-oic acid (F2-3) [21], 3-*O*-*cis*-caffeoyl-oleanolic acid (F4-1), 3-*O*-*trans*-caffeoyl-oleanolic acid (F4-2) [22], and 3-*O*-*trans*-*p*-coumaroyl-oleanolic acid (F7-1) [23] (Figure 9 and Appendix A). Unfortunately, F2-4 could not be precisely identified due to the lack of a mass response and poor NMR resolution. It was not a triterpenoid and seemed to be a flavonoid due to a positive response to Neu’s reagent.

### 2.9. Cytotoxicity and Antiviral Activity of the Purified Compounds on HCoV-229E and SARS-CoV-2

To determine whether the purified compounds were responsible for the antiviral effects on both coronaviruses, we conducted cytotoxicity and antiviral dose-response experiments. Huh-7 cells were treated with the various compounds at different concentrations of up to 100 µM for 24 h, and the cytotoxicity was quantified using an MTS assay. Among the compounds, F4-2 displayed the highest CC_50_ value of 81.4 µM. Conversely, the mixture F4-1 exhibited the lowest CC_50_ value of 21 µM. The CC_50_ values for F2-2 and F2-3 were found to be 44.4 and 52.1 µM, respectively. Both F2-1 and F7-1 demonstrated similar CC_50_ values of 39.9 µM and 39.3 µM, respectively (Table 4, Appendix A). 

Antiviral testing against HCoV-229E showed that the six compounds isolated from the three fractions (F2, F4, and F7) demonstrated a dose-dependent antiviral activity in both Huh-7 and Huh-7/TMPRSS2 cells (Table 4, Appendix A). F2-1 displayed the most promising performance, showing IC_50_ values of 8.6 and 9.1 µM in Huh-7 and Huh-7/TMPRSS2 cells, respectively, with an SI of approximately 4. It is noteworthy that the major compound F2-4 of fraction 2, which was not chemically characterized, did not exhibit either cytotoxicity or antiviral effects on HCoV-229E (Appendix A). F4-2 also displayed interesting antiviral activities and the highest SI of 7. F4-1 was the most active but only in Huh-7 cells with an IC_50_ value of 7.6 µM.

Finally, we explored the antiviral impact of the three cinnamoyl oleanolic acids, F4-1, F4-2, and F7-1, on SARS-CoV-2 at various concentrations. The inhibitory activity was assessed by quantifying the levels of N protein expression (Figure 10). A noticeable, dose-dependent reduction in N protein levels, indicative of an antiviral effect, was observed for all three compounds. Specifically, F4-1 exhibited the highest antiviral activity, significantly inhibiting SARS-CoV-2 by 54% at 12.5 µM, 68% at 25 µM, and 93.3% at 50 µM. In contrast, both F4-2 and F7-1 demonstrated significant inhibitory effects, reaching 54% and 55.5%, respectively, but only at the concentration of 50 µM (Figure 10). The difference in activity observed between F4-1 and F4-2 could suggest better activity for 3-*O*-*cis*-caffeoyl oleanolic acid. Unfortunately, this compound could not be obtained in pure form due to a spontaneous conversion to the *trans* isomer during the purification process.

Taken together, the results show that several cinnamoyl triterpenoids, isolated from three different fractions of HR DCM SE, have antiviral activities against both HCoV-229E and SARS-CoV-2.

## 3. Discussion

In light of the emergence of coronavirus outbreaks and the predictability of future epidemics and pandemics, it has become imperative to discover effective antiviral solutions. Despite the development of vaccines and antiviral drugs, several significant challenges hinder progress, such as unequal access to treatments and vaccines, the emergence of variant strains, and more. Therefore, there is a critical need for effective, accessible, and cost-effective antiviral treatments targeting SARS-CoV-2, especially in low-income countries.

Natural products have played a crucial role in the field of drug discovery, particularly in the discovery of antibacterial and antitumoral agents. Halophytes and salt-tolerant plants are recognized as abundant sources of specialized metabolites that exhibit a wide range of biological functions, including survival under challenging environmental conditions and defense mechanisms against microorganisms. In our study, we explored the antiviral potential of halophytes and salt-tolerant plants collected from the North Sea and English Channel coasts in northern France against various coronaviruses.

The crude methanolic extracts of *Hippophae rhamnoides*, *Salix repens* (S and R), and *Baccharis halimifolia* exhibited significant antiviral effects against HCoV-229E and SARS-CoV-2. A recent paper demonstrated the antiviral activity of *Salix* spp against both seasonal and pandemic coronaviruses; however [24]. Furthermore, a flavonoid compound, isorhamnetin, isolated from *Hippophae rhamnoides* fruits, displayed antiviral activity against the SARS-CoV-2 spike pseudotyped virus in vitro [25]. However, no prior research has investigated the antiviral activity of these plants against HCoV-229E.

*Hippophae rhamnoides* belongs to the Elaeagnaceae family and is native to the cold-temperate regions of Europe and Asia [26]. Sea buckthorn berries are known for their rich nutritional content, including vitamins and specialized metabolites like tocopherols, phenolic acids, carotenoids, flavonoids, tocopherols, and phytosterols [27,28]. Traditional medicine in China and Russia has previously employed this plant to treat dermatological diseases. Additionally, numerous studies have highlighted the pharmacological effects of *Hippophae rhamnoides*, including its antioxidant [29], antimicrobial [30], anti-atherogenic [31], cardioprotective [31], hepatoprotective [32], radioprotective [33], and tissue regeneration properties [34]. Nonetheless, research on the antiviral activity of *Hippophae rhamnoides* remains limited.

Liquid–liquid partitioning, combined with simultaneous biological testing, revealed that the HR DCM SE exhibited the most substantial antiviral effect against coronaviruses. Consequently, we decided to narrow our investigation to this specific sub-extract to pinpoint the active fraction and subsequently isolate the bioactive compounds responsible for the antiviral effect against coronaviruses. The nonpolar nature of this sub-extract suggests that the active compounds are rather lipophilic.

Using a bioguided fractionation approach combining fractionation by CPC and antiviral testing, we identified F3, F4, and F7 as the most potent fractions against HCoV-229E, whereas F2, F3, F4, and F7 were the most effective against SARS-CoV-2. UHPLC-UV-MS analysis revealed the presence of compounds, with a similar absorbance and molecular mass of around 600 g.mol^−1^ found in several fractions, suggesting that analogs contribute to the antiviral activity in different fractions.

Six compounds were isolated from HR DCM SE and identified through NMR and HR-MS analysis as cinnamoyl triterpenoids. Among these, three compounds—2-*O*-*trans-p*-coumaroyl-maslinic acid, 3β-hydroxy-2α-*trans*-*p*-coumaryloxy-urs-12-en-28-oic acid, and 3β-hydroxy-2α-*cis*-*p*-coumaryloxy-urs-12-en-28-oic acid—were isolated from F2 and are derivatives of maslinic acid (MA) and ursolic acid (UA), respectively. The remaining compounds were cinnamoyl derivatives of oleanolic acid (OA), obtained from F4 (mixture of 3-*O*-*trans*-caffeoyl oleanolic acid/3-*O*-*cis*-caffeoyl oleanolic acid (70/30) and 3-*O*-*trans*-caffeoyl oleanolic acid) and F7 (3-*O*-*trans-p*-coumaroyl oleanolic acid). MA, UA, and OA are common triterpenoids and are known to be abundant in *Hippophae rhamnoides* [28].

Despite F2 not demonstrating an antiviral effect against HCoV-229E, the UHPLC-UV-MS analysis indicated the presence of a major compound, seemingly a flavonoid, which also proved to be inactive when tested at different doses. However, the antiviral impact of F2 on SARS-CoV-2 prompted us to isolate three other compounds, which are cinnamoyl derivatives of MA and UA. Unexpectedly, these compounds exhibited inhibitory activity against HCoV-229E following separation and isolation, whereas F2 was inactive. This might be due to the fact that these three compounds were present at low levels in F2.

Triterpenoids represent the most widely distributed category of natural compounds, typically originating from the C30 molecular structure, which is synthesized by rearranging six isoprene units following the isoprene rule. They are found in plants either in their free form or as glycosides (saponins). Some of these triterpenoids can, in some cases, be acylated. Among triterpenoids, the tricyclic and pentacyclic varieties are the most abundant [32,33]. Previous phytochemical studies conducted on different parts of *Hippophae rhamnoides* highlighted the presence of pentacyclic triterpenoids, mostly the oleanane and ursane types, with different biological activities [28]. Some cinnamoyl triterpenoids, including 2-*O*-*trans*-p-coumaroyl maslinic acid, 2-*O*-*trans*caffeoyl maslinic acid, 3-*O*-*trans*-*p*-coumaroyl oleanolic acid, and 3-*O*-*trans*-caffeoyl oleanolic acid have already been isolated from the branch bark of this plant [20]. Some other derivatives have been tentatively identified in different parts of *Hippophae rhamnoides* by LC-HRMS [35].

Pentacyclic triterpenes, such as analogs or derivatives of OA, have demonstrated various inhibitory activities against viruses, primarily linked to their structures. They have been found to be effective against the influenza virus and hepatitis C virus (HCV) infections. These triterpenes work by binding to viral fusion proteins like hemagglutinin (HA2) of influenza [36], E2 of HCV [37], and GP41 of human immunodeficiency virus-1, thus disrupting the entry of the virus into host cells. Further exploration of OA revealed that it can interact with heptad repeat-2 and hinder Ebola virus–cell fusion [38], shedding light on its mechanism of action against SARS-CoV-2. Additionally, research suggests that OA may act as an inhibitor of viral replication by blocking the activity of SARS-CoV 3CLpro [39]. Moreover, friedlane-type triterpenoids isolated from *Euphorbia neriifolia* L. leaves, a drought-tolerant plant, exhibited potent antiviral activity against HCoV-229E cultured in MRC-5 cells [40]. The friedelane skeleton could act on multiple targets simultaneously, making it a potential candidate for exerting an antiviral effect against various human coronaviruses.

MA is a commonly occurring triterpenoid abundant in the fruits of *Hippophae rhamnoides* [41]. MA has demonstrated a wide spectrum of biological activities, including antibacterial, anti-inflammatory, and antitumor properties. Moreover, in another study, a cinnamoyl maslinic acid named 3-β -O-(*trans*-*p*-coumaroyl)maslinic acid, demonstrated broad antimicrobial activity against Gram-positive bacteria and yeasts, with a minimum inhibitory concentration of 12.5 µg/mL against *Staphylococcus capitis* and *Candida albicans* [42]. 

UA and OA share similar chemical structures but differ in the position of one methyl group on ring E. UA has been recognized for its anti-inflammatory, antibacterial, antioxidant, anti-diabetic, and anticancer properties. While there are limited data regarding the antiviral activity of UA against HCoV-229E, there are several reports related to its potential against SARS-CoV-2, given their similar morphologies, replication cycles, and symptoms. It has been tested against the SARS-CoV-2 Mpro enzyme and successfully inhibited its activity [43]. Additionally, molecular docking (MD) and molecular dynamic simulation studies have confirmed the ability of UA and its derivatives to interact with SARS-CoV-2 protease during 50 nanoseconds of MD simulation [44]. UA exhibits high binding affinity, forming a hydrogen bond with the amino group of Asp 108 in the PLpro protease enzyme and engaging in hydrophobic interactions with Ala 107, Pro 248, and Tyr 264 of the same enzyme [45]. In silico studies suggest that UA could inhibit the interaction between SARS-CoV-2 spike proteins and the receptor, angiotensin-converting enzyme 2 (ACE-2) [46]. However, further confirmation is needed through in vitro or in vivo studies. 

In this study, we showed that the antiviral activity of cinnamoyl terpenoids on human coronaviruses is promising. However, they display relatively low SIs (between 1 and 7) due to their cytotoxicity. The toxicity and antiviral assays were performed in Huh-7 cells, which is a hepatoma cell line. It would be interesting to evaluate the cytotoxicity of the compounds in respiratory cell lines or in animal models (in vivo). Few studies exist on this type of compound. Furthermore, if the cytotoxicity is high in other cell lines, one could envisage administrating the compound via aerosols (orally or nasally), limiting any toxic side effects. This mode of administration would reach the nasal or bronchial epithelial cells, the sites of viral replication.

Additional work is necessary to determine the mechanism of action of the active isolated compounds against SARS-CoV-2 and HCoV-229E. It would be necessary to determine if they act in the entry or replication step. It is unlikely that one compound could act in both steps; however, it would be very interesting to show that *Hippophae rhamnoides* extract contains a mixture of compounds, with some active on entry and others on replication. It would also be interesting to perform combination assays with the different compounds to determine if the mixture of all these compounds is more active than each isolated molecule.

Considering all the aforementioned information, it is evident that pentacyclic triterpenoids exhibit anti-coronavirus activity, perhaps due to their structural properties. The presence of cinnamoyl triterpenoid derivatives like MA, UA, and OA, which share structural similarities, in plants like *Hippophae rhamnoides* points to potential antiviral activity, particularly against SARS-CoV-2. Further in-depth investigations are necessary to gain a deeper understanding of their specific targets and mechanisms of action, facilitating their development and ensuring safety, and fully exploring their therapeutic efficacy.

## 4. Materials and Methods

### 4.1. Plant Material

Plant species, mainly halophytes, were selected and collected between July 2020 and November 2020 from five different locations (Étaples, Dannes, Le Portel, Gravelines, Zuydcoote) distributed across the coastline region of northern France (Hauts-de-France region) in conjunction with the managers of the natural sites (Figure 11). Plants were mainly collected from schorres; coastal cliffs; and incipient, established, and relict dunes [46,47,48]. All these operations, followed by the identification of plant materials, were conducted by Prof. Céline Rivière and Dr. Gabriel Lefèvre from the Faculty of Pharmacy in Lille (UMRt BioEcoAgro). These samples were collected in accordance with the rules of the Nagoya Protocol and the French biodiversity law of 2017 (decision of 23 September 2020 issued by the Ministry of Ecological and Inclusive Transition; NOR: TREL2002508 S/342 and ABSCH-IRCC-FR-252501-1). Specific authorizations were also granted by the “Direction Interrégionale de la Mer Manche Est- Mer du Nord” (Decision n°778/2020) and by the prefect of the region of Normandy (regulation service for maritime activities). Harvested plant species were dried at 30 °C in an oven for a maximum of one week and protected from light. Different parts of these plants (leaves, stems, roots) were pulverized separately using a crushed Retsh Cutting Mill SM200.

### 4.2. Solid/Liquid Extraction

Crude methanolic extracts were prepared by maceration by soaking the powder in 10 mL/g of methanol for 24 h, and the mixtures were then filtered through Whatman filter paper (11 μm pore size). The process was repeated three times. The resulting extracts were then dried in vacuum at 35 °C using a rotary evaporator and stored at −20 °C until tested. For cytotoxicity and antiviral assays, extracts were re-suspended in DMSO at 25 mg/mL, aliquoted, and stored at −20 °C.

### 4.3. Liquid/Liquid Extraction

The active crude methanolic extracts were subjected to bioguided fractionation using liquid–liquid partitioning. An amount of 3 g of crude methanolic extract was dissolved in water and then partitioned with DCM and EtOAc (3 × 300 mL) to obtain three solvent partitions. The partitions were evaporated using a rotary evaporator and transferred to vials for storage. The apolar partitions (DCM and EtOAc) were evaporated at an ambient temperature and then desiccated under vacuum (desiccator); the polar (Aq) partitions were dried by lyophilization (freeze-drying). 

### 4.4. Fractionation of the DCM Sub-Extract of Hippophae Rhamnoides

The DCM extract of *Hippophae rhamnoides* (HR DMC SE) was fractionated by centrifugal partition chromatography (Armen instruments^®^, Saint-Avé, France) with a capacity of 1 L. The liquid phases were pumped with a Shimazu^®^ pump (LC-20AP, Kyoto, Japan). The column was coupled online with a DAD detector (SPD-M20A). Fractions were collected with an automated fraction collector (Gilson^®^ FC 204, Villiers-le-Bel, France). The elution profile was recorded using LabSolutions™ software version 1.25. The solvent system was composed of heptane, EtOAc, methanol, and water (2:1:2:1) (Arizona R system). The CPC rotor was first filled with the stationary phase (upper phase) at a flow rate of 50 mL.min^−1^ (500 rpm) in descending mode. Equilibrium was reached by introducing the mobile phase (lower phase) at 1200 rpm and a flow rate of 30 mL.min^−1^. An amount of 5.9 g of the HR DCM SE (obtained from the whole plant) was dissolved in 46 mL of the organic/aqueous phase mixture (1:1, *v*/*v*) and filtered through a Millipore syringe filter (0.45 μm). The filtered solution was injected immediately after displacement of the stationary phase (170 mL). The elution was carried out at 30 mL.min^−1^ for 60 min and monitored at λ = 254 nm. After that, extrusion mode was performed to allow for the recovery of highly retained molecules in the stationary phase. At the end of the CPC cycle, the 193 tubes obtained were characterized by UHPLC-UV-MS and then grouped into 10 fractions according to their phytochemical profiles. The 10 fractions were then concentrated by a centrifugal concentrator (Genevac™, Fisher Scientific, Illkirch, France).

### 4.5. UHPLC-UV-MS Analysis 

The Acquity UPLC H-Class Waters^®^ System (Guyancourt, France) apparatus was equipped with two independent pumps, a controller, a diode array detector (DAD), and a QDa electrospray quadrupole mass spectrometer. The stationary phase was a C18 BEH (2.1 × 50 mm, 1.7 µm) reverse column. The mobile phase was composed of two solvents: (A) ultrapure water + 0.1% formic acid (Carlo Erba Reagents^®^, Val de Reuil, France) and (B) Acetonitrile (Carlo Erba Reagents^®^, Val de Reuil, France) + 0.1% formic acid. The flow rate and column temperature were set at 0.3 mL.min^−1^ and 30 °C, respectively. The wavelength range was fixed at 200–790 nm with a resolution of 1.2 nm. Ionization was carried out in both negative and positive modes, with the mass ranging from 50 to 1250 Da. The cone voltage and capillary voltage values were 15 V and 0.8 kV, respectively. The injection volume was set at 2 µL. UHPLC-UV-MS analysis was executed following the elution program: 10%→100% (B) (0–9 min), 100% (B) (9–11.5 min), and 10% (B) (11.5–14 min). All samples were prepared at 1 mg·mL^−1^ in analytical grade MeOH and filtered through a PTFE 0.4 µm membrane before injection. 

### 4.6. Preparative HPLC 

The equipment consisted of Shimadzu^®^ LC-20AP binary high-pressure pumps, an SPD-M20A photodiode array detector, and a CBM-20A controller. A column Interchim US5C18HQ-250/212 Uptisphere Strategy C18-HQ 5 µm (250 × 21.2 mm) prep-LC was used in this experiment as the stationary phase. The mobile phase was composed of ultra-pure water (Millipore Integral 5 Milli-Q, Merck™, Trosly-Breuil, France) + 0.1% formic acid (Merck™, Darmstadt, Germany) (solvent A) and acetonitrile (Carlo Erba Reagents^®^, Val de Reuil, France) (solvent B). The flow rate was set at 15 mL·min^−1^. The purification monitoring was carried out at two main wavelengths: 247 nm and 254 nm. Preparative HPLC was performed on fractions F2, F4, and F7 obtained from HR DCM SE after CPC. The gradients used were 60% B (0.01 min), 60% to 100% B (0.01–25 min), 100% B (25.01–28.99 min), and 60% B (29–30 min) for fraction 2, and 80% B (0.01 min), 80% to 100% B (0.01–25 min), 100% B (25.01–28.99 min), and 80% B (29–30 min) for fractions 4 and 7.

### 4.7. NMR and HRMS

The structures of the purified compounds were determined using NMR and HR-MS. NMR spectra (mono- and bi-dimensional) were recorded on a Bruker ^®^ DPX-500 spectrometer (^1^H- and ^13^C-NMR at 500 and 125 MHz) (Bruker, Bremen, Germany). High-Resolution Mass Spectrometry (HR-MS) analyses were carried out using a Thermo Fisher Scientific^®^ Exactive Orbitrap Mass Spectrometer (Thermo Fisher Scientific, Waltham, MA, USA) equipped with an electrospray ion source. The pure compounds were analyzed in deuterated methanol, MeOD (Euriso-Top^®^, Gif-sur-Yvette, France). HR-MS analyses were carried out in negative mode with a range of *m*/*z* 100–1000 amu. Products were solubilized in methanol.

### 4.8. Virus and Cell Lines

The human hepatoma cell line (Huh-7), whether expressing the TMPRSS2 protease or not [18], and the African green monkey kidney Vero-81 cells were grown in DMEM supplemented with GlutaMax-I and 10% fetal bovine serum and cultured at 37 °C in 5% CO_2_ in a humidified incubator. All cell lines used in this study were regularly screened for mycoplasma contamination using the MycoAlert^TM^ Mycoplasma Detection Kit (Lonza Bioscience, Basel, Switzerland).

The viruses used were HCoV-229E strain VR-740 (ATCC), a recombinant HCoV-229E-Luc (kind gift of Volker Thiel), and SARS-CoV-2 (isolate SARS-CoV- 536 2/human/FRA/Lille_Vero-81-TMPRSS2/2020, NCBI MW575140). 

### 4.9. Cell Viability Assay

Huh-7 cells and Vero-81 cells were seeded in 96-well plates and incubated with 100 μL of culture medium containing increasing concentrations of our compounds for 24 h. An MTS based viability assay (CellTiter 96 aqueous nonradioactive cell proliferation assay, Promega, Madison WI, USA) was performed, as recommended by the manufacturer. The absorbance of formazan at 490 nm was detected using ELx808 plate reader (BioTek Instruments Inc., Winooski, VT, USA). Each measurement was performed in triplicate.

### 4.10. Virus Infection Assay

#### 4.10.1. HCoV-229E

Huh-7 and Huh-7/TMPRSS2 cells, were seeded in 96-well plates and inoculated with HCoV-229E-Luc at an MOI of 0.3 simultaneously with the compounds for 7 h and then lysed in 20 μL of 1× luciferase lysis buffer (Promega) as described [49]. The luciferase activity was quantified in a TriStar LB 941 luminometer (Berthold Technologies, Bad Wildbad, Germany) using the Renilla luciferase assay system (Promega), as recommended by the manufacturer.

#### 4.10.2. SARS-CoV-2

Vero-81 cells were seeded in 24-well plates, inoculated with SARS-CoV-2, and incubated simultaneously with the different compounds for 16 h [50]. Cells were lysed in ice-cold lysis buffer (Tris HCl, 50 mM; NaCl, 100 mM; EDTA, 2 mM; Triton X-100, 1%; SDS, 0.1%) on ice for 20 min. Lysates were collected and analyzed by Western blotting using rabbit polyclonal anti-SARS-CoV-2 nucleocapsid antibodies (Novus Biologicals, Littleton, CO, USA) and mouse anti-β-tubulin monoclonal antibody (TUB 2.1) from Sigma. Horse-radish peroxidase-conjugated goat anti-rabbit and anti-mouse secondary antibodies (Jackson ImmunoResearch, West Grove, PA, USA) were used for the revelation using an enhanced chemiluminescence (ECL) Western blotting substrate (Thermo Fisher Scientific). The intensity of the bands was quantified using ImageJ software version 1.53i.

#### 4.10.3. Statistical Analysis

The results were presented as the means ± SEM of three independent experiments performed in triplicate. The data were analyzed using GraphPad Prism software version 10.0.3 (Boston, MA, USA) by comparing each treated group and untreated group (DMSO control).

## Figures and Tables

**Figure 1 ijms-24-16617-f001:**
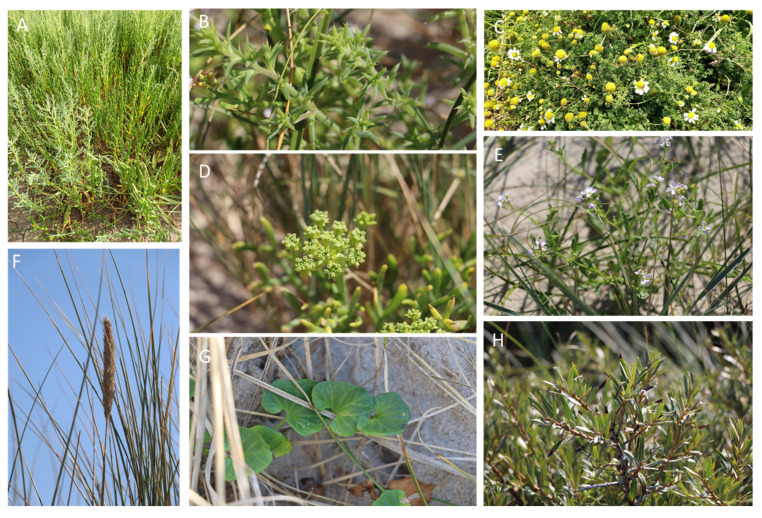
Pictures of some collected plant species (Lefèvre G. and Rivière C., 2020): (**A**) *Salicornia procumbens* Sm. and *Suaeda maritima* (L.) Dumort. (Etaples), (**B**) *Salsola kali* L. (Dannes), (**C**) *Tripleurospermum maritimum* (L.) W. D. J. Koch (Le Portel), (**D**) *Crithmum maritimum* L. (Le Portel), (**E**) *Cakile maritima* Scop. subsp. *integrifolia* (Hornem.) Greuter and Burdet (Dannes), (**F**) *Ammophila arenaria* subsp. *arenaria* (L.) Link (Dannes), (**G**) *Convolvulus soldanella* L. (Dannes), and (**H**) *Hippophae rhamnoides* L. (Dannes).

**Figure 2 ijms-24-16617-f002:**
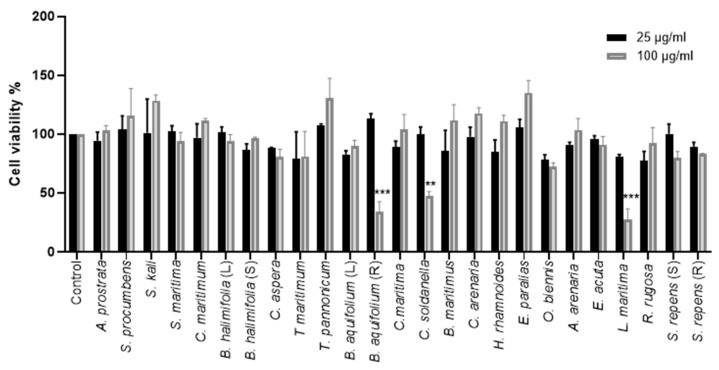
Cytotoxicity of the methanolic crude extracts on Huh-7 cells. Huh-7 cells were treated with two concentrations, 25 and 100 μg/mL, of crude extracts. Control cells were treated with 0.1% DMSO only. The cells were incubated for 24 h, and an MTS assay was then performed to determine the cell viability. The data bars represent the mean ± standard error of the mean (SEM) of three experiments performed in triplicate. The asterisk indicates a statistical difference compared to the control. (**, *p* < 0.01; ***, *p* < 0.001). (L) = Leaves, (S) = Stem, (R) = Roots.

**Figure 3 ijms-24-16617-f003:**
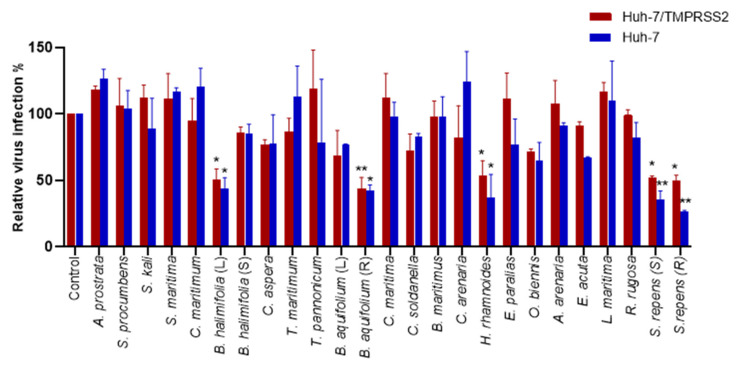
Screening of the antiviral activity of crude methanolic extracts on HCoV-229E-Luc. Huh-7 or Huh-7/TMPRSS2 cells were inoculated with HCoV-229E-Luc in the presence of various plant extracts at 25 µg/mL. Cells were lysed 7 h post-inoculation, and luciferase activity was quantified. Experiments were performed in triplicate, with each experiment being repeated thrice. The data bars represent the mean ± SEM. The asterisk indicates a statistical difference compared to the control (*, *p* < 0.05; **, *p* < 0.01).

**Figure 4 ijms-24-16617-f004:**
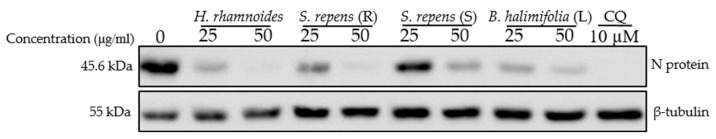
Antiviral activity on SARS-CoV-2 of crude methanolic extract of *Hippophae rhamnoides*, *Salix repens* (R), *Salix repens* (S), and *Baccharis halimifolia* (L). Vero-81 cells were infected with SARS-CoV-2 in the presence of different plant extracts at 25 and 50 µg/mL, or 10 µM chloroquine (CQ). Cell lysates were collected after 16 h and subjected to immunoblotting analysis using an anti-SARS-CoV-2 N antibody and an anti-ß-tubulin antibody to show an equal amount of cellular protein in each lane. This immunoblot is representative of two independent experiments.

**Figure 5 ijms-24-16617-f005:**
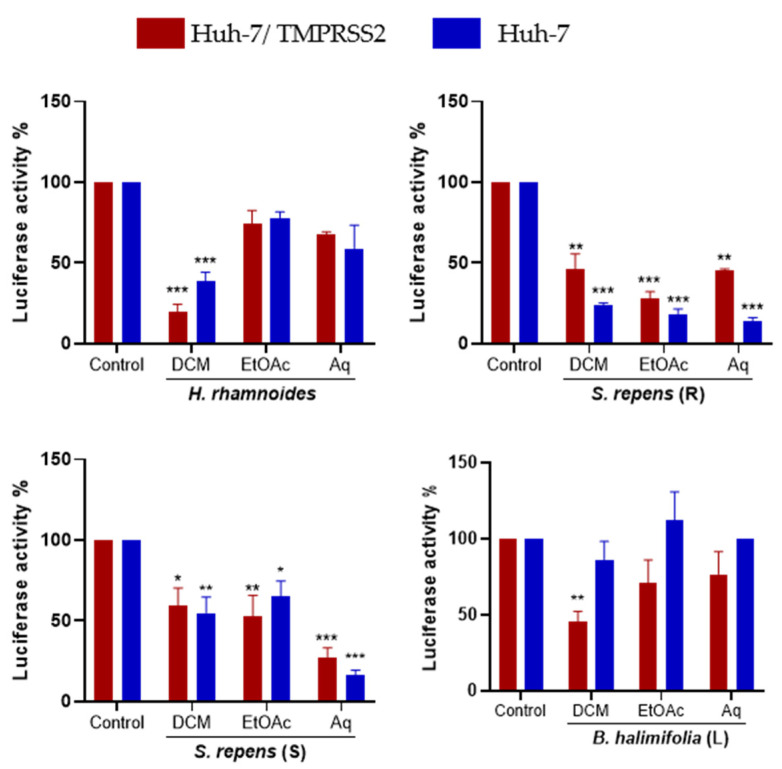
Inhibitory activity of the three partitions obtained from each plant’s crude methanolic extract on HCoV-229E-Luc infection. Experiments were conducted as described earlier. Data are represented as the mean ± SEM of three independent experiments. (*, *p* < 0.05; **, *p* < 0.01; ***, *p* < 0.001).

**Figure 6 ijms-24-16617-f006:**
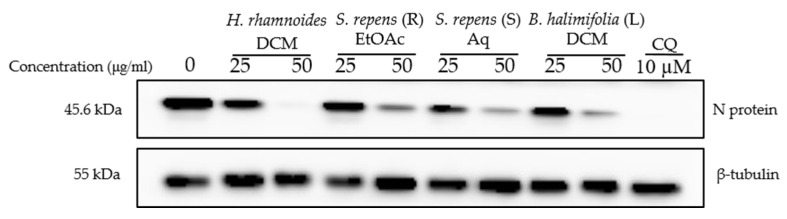
Antiviral activity on SARS-CoV-2 of sub-extracts. Vero-81 cells were infected with SARS-CoV-2 in the presence and absence of different plant sub-extracts at 25 and 50 µg/mL, or 10 µM chloroquine. Cell lysates were collected after 16 h and subjected to immunoblotting as described. This blot is representative of two independent experiments.

**Figure 7 ijms-24-16617-f007:**
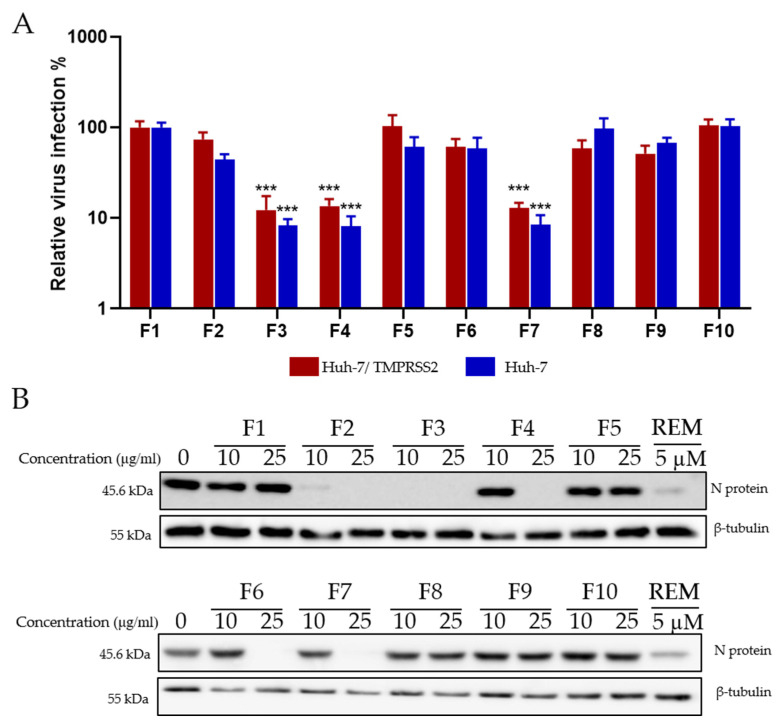
Antiviral activity of *Hippophae rhamnoides* fractions. (**A**) Screening of the antiviral activity of the fractions of *Hippophae rhamnoides* with HCoV-229E-Luc. The different fractions were tested at 25 μg/mL. (**B**) Western blotting analysis showing the effects of the different fractions of *Hippophae rhamnoides* on N protein expression in Vero-81 cells. Vero-81 cells were infected with SARS-CoV-2 in the presence of the different fractions at 10 and 25 µg/mL. Cell lysates were collected after 16 h and subjected to Western blotting. (***, *p* < 0.001).

**Figure 8 ijms-24-16617-f008:**
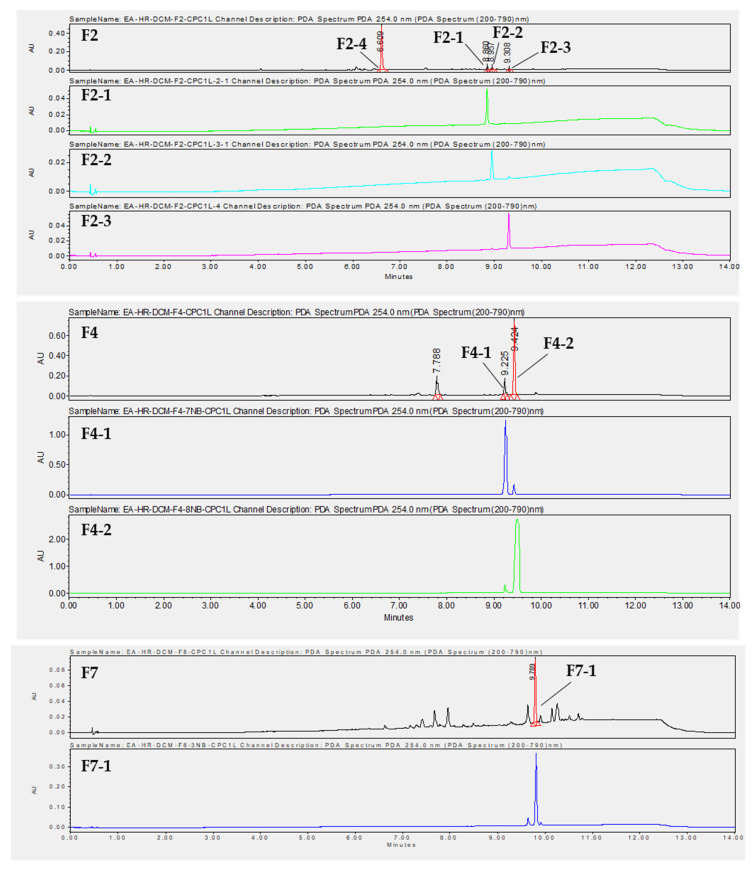
Chromatograms at λ = 254 nm obtained by UHPLC-UV-MS of F2, F4, and F7, as well as purified compounds obtained by preparative HPLC.

**Figure 9 ijms-24-16617-f009:**
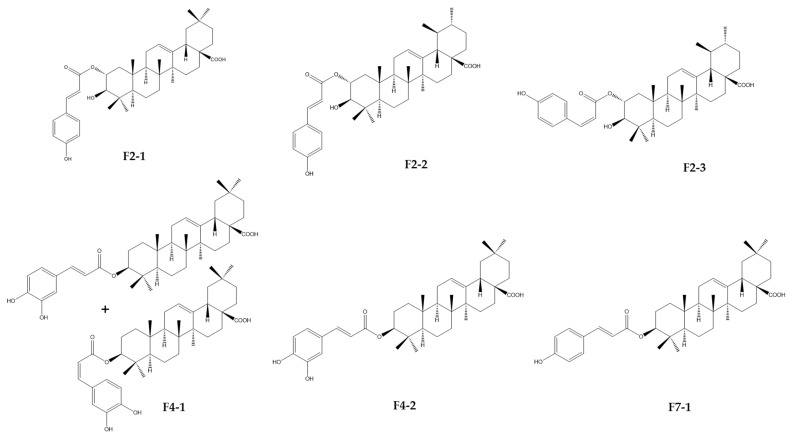
Chemical structures of cinnamoyl triterpenoids purified from fractions F2, F4, and F7 resulting from CPC fractionation of the DCM sub-extract of *Hippophae rhamnoides*.

**Figure 10 ijms-24-16617-f010:**
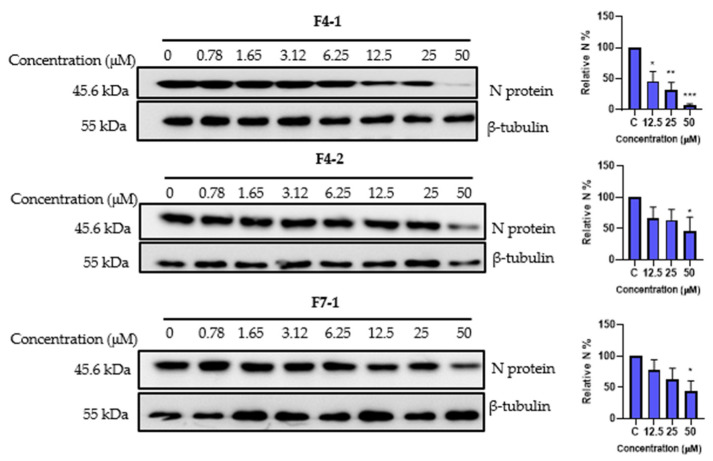
Antiviral activity of cinnamoyl oleanolic acids against SARS-CoV-2. The antiviral effect of F4-1 (3-O-cis-caffeoyl-oleanolic acid), F4-2 (3-O-trans-caffeoyl-oleanolic), and F7-1 (3-O-trans-p-coumaroyl-oleanolic acid) against SARS-CoV-2 was determined by Western blot. Vero-81 cells were treated with the different compounds at different concentrations for 16 h. Cell lysates were collected after 16 h and subjected to Western blotting. Data values represent the mean ± standard deviation from 3 independent experiments, (*, *p* < 0.05; **, *p* < 0.01; ***, *p* < 0.001).

**Figure 11 ijms-24-16617-f011:**
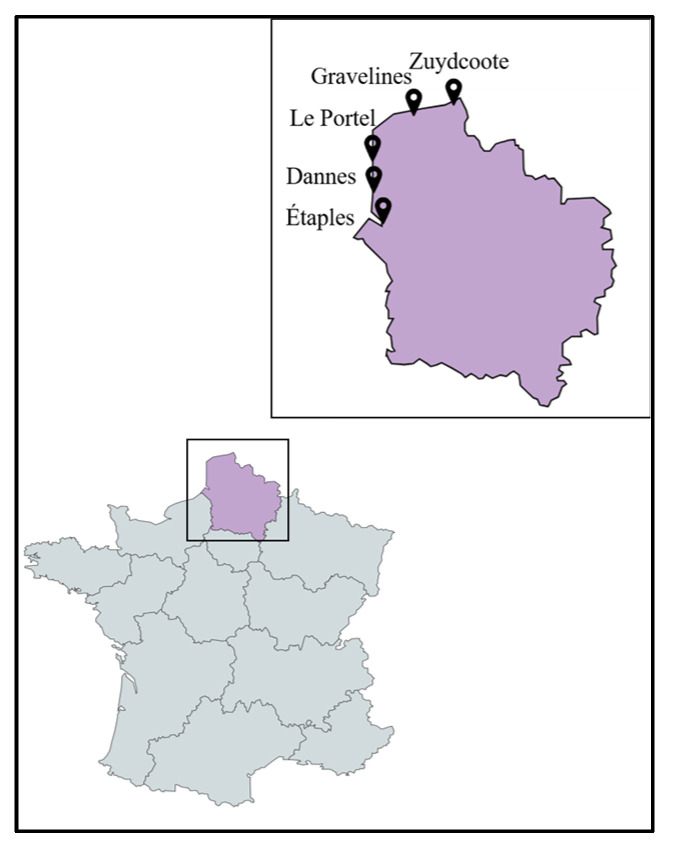
Map showing the sites of collection of the studied plant species.

**Table 1 ijms-24-16617-t001:** List of plant species used in this study.

Botanical Family	Plant Species	Parts	Place of Collection	Voucher Code
Amaranthaceae	*Atriplex prostrata* Boucher ex DC.	Whole plant	Dannes (at the base of an incipient dune)	LIP007584
	*Salicornia procumbens* Sm.	Whole plant	Etaples (in the schorre)	LIP007585
	*Salsola kali* L.	Whole plant	Dannes (at the base of an incipient dune)	LIP007586
	*Suaeda maritima* (L.) Dumort.	Whole plant	Etaples (in the schorre)	LIP007587
Apiaceae	*Crithmum maritimum* L.	Aerial parts	Le Portel (on a coastal cliff)	LIP007588
	*Baccharis halimifolia* L.	Leaves (L), stems (S)	Gravelines (planted in a roadside hedge)	LIP007589
Asteraceae	*Centaurea aspera* L.	Whole plant	Zuydcoote (on a relict foredune)	LIP007590
	*Tripleurospermum maritimum* (L.) W. D. J. Koch	Whole plant	Le Portel (on a coastal cliff)	LIP007591
	*Tripolium pannonicum* (Jacq.) Dobroc.	Whole plant	Dannes (in the schorre)	LIP007592
Berberidaceae	*Berberis aquifolium* Pursh	Leaves (L), roots (R)	Zuydcoote (on a relict foredune)	LIP007593
Brassicaceae	*Cakile maritima* Scop. subsp. *integrifolia* (Hornem.) Greuter and Burdet	Whole plant	Dannes (at the base of an incipient dune)	LIP007594
Convolvulaceae	*Convolvulus soldanella* L.	Whole plant	Dannes (on an incipient dune)	LIP007595
Cyperaceae	*Bolboschoenus maritimus* (L.) Palla	Whole plant	Dannes (in the schorre)	LIP007596
	*Carex arenaria* L.	Whole plant	Dannes (in the schorre)	LIP007597
Elaeagnaceae	*Hippophae rhamnoides* L.	Whole plant	Dannes (on an established foredune)	LIP007598
Euphorbiaceae	*Euphorbia paralias* L.	Whole plant	Dannes (on an incipient foredune)	LIP007599
Onagraceae	*Oenothera biennis* L.	Whole plant	Zuydcoote (on a relict foredune)	LIP007600
Poaceae	*Ammophila arenaria* subsp. *arenaria* (L.) Link	Whole plant	Dannes (on an established foredune)	LIP007601
	*Elytrigia acuta* (DC.) Tzvelev	Whole plant	Zuydcoote (on an incipient foredune)	LIP007602
Primulaceae	*Lysimachia maritima* (L.) Galasso, Banfi, and Soldano	Whole plant	Dannes (in the schorre)	LIP007603
Rosaceae	*Rosa rugosa* Thumb.	Aerial parts	Dannes (on an established foredune)	LIP007604
Salicaceae	*Salix repens* subsp. *dunensis* Rouy	Stems (S), roots (R)	Zuydcoote (on a relict foredune)	LIP007605

**Table 2 ijms-24-16617-t002:** Cytotoxicity, antiviral activity, and SI of each of the crude methanolic extracts against HCoV-229E.

Crude Extract	CC_50_ (μg/mL) Vero-81	CC_50_ (μg/mL) Huh-7	Huh-7		Huh-7/TMPRSS2	
			IC_50_ (μg/mL)	SI	IC_50_ (μg/mL)	SI
*H. rhamnoides*	499	621	19.7	31.5	27.2	22
*S. repens (R)*	441	149	29.1	5.1	15.5	9
*S. repens (S)*	438	285	15.1	28.9	14.7	19
*B halimifolia (L)*	820	>1000	65.3	>15	11.2	>89

**Table 3 ijms-24-16617-t003:** Cytotoxicity, activity, and SI of each of the sub-extracts against HCoV-229E.

Sub-Extract	Vero-81 CC_50_ (μg/mL)	Huh-7 CC_50_ (μg/mL)	Huh-7		Huh-7/TMPRSS2	
			IC_50_ (μg/mL)	SI	IC_50_ (μg/mL)	SI
*H. rhamnoides* DCM	264	410	18.7	21	36.	11
*S. repens* (R) EtOAc	344	500	15.8	31	28.87	17
*S. repens* (S) Aq	262	550	7.6	71	30.5	18
*B. halimifolia* (L) DCM	347	368	31.1	11	38.3	9

**Table 4 ijms-24-16617-t004:** Cytotoxicity, antiviral activity, and SI of each purified compound against HCoV-229E.

Sub-Extract	Huh-7 CC_50_ (µM)	Huh-7	Huh-7/TMPRSS2		
		IC_50_ (μM)	SI	IC_50_ (μM)	SI
**F2-1** (2-*O*-*trans*-*p*-coumaroyl-maslinic acid)	39.9	8.6	4	9.1	4
**F2-2** (3β-hydroxy-2α-*trans*-*p*-coumaryloxy-urs-12-en-28-oic acid)	44.4	12.0	3	11.4	3
**F2-3** (3β-hydroxy-2α-*cis*-*p*-coumaryloxy-urs-12-en-28-oic acid)	52.1	14.5	3	14.1	3
**F4-1** Mixture 3-*O*-*trans*-caffeoyl oleanolic acid / 3-*O*-*cis*-caffeoyl oleanolic acid (70/30)	21.0	7.6	2	12.0	1
**F4-2** (3-*O*-*trans*-caffeoyl-oleanolic acid)	81.4	11.6	6	11.5	7
**F7-1** (3-*O*-*trans*-*p*-coumaroyl-oleanolic acid)	39.3	11.4	3	10.7	3

## Data Availability

Data are contained within the article and Appendix A. Raw data are available on request from the corresponding author.

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
