# Peer review of "Discovery of Anti-Coronavirus Cinnamoyl Triterpenoids Isolated from Hippophae rhamnoides during a Screening of Halophytes from the North Sea and Channel Coasts in Northern France"

_ijms, 2023, doi:10.3390/ijms242316617_

Round 1

Reviewer 1 Report

Comments and Suggestions for Authors

This work focuses on the discovery of antiviral agents against coronaviruses from halophytes and involves the isolation and identification of their antivirals. The presented data has the potential to be valuable for the development of new antiviral drugs. However, there are some specific suggestions I would like to make to enhance the quality of the article

Introduction

Line 54- 56: “…only two drugs, paxlovid (nirmatrelvir and ritonavir) and molnupiravir, have been granted authorization by the Food and Drug Administration (FDA).” This information does not appear to be supported by the reference provided (Reference [4]). Please ensure that the source clearly supports the statement.

Line 56: Please provide the full name of EMA (European Medicines Agency).

Line 73: Edit this to “salt (NaCl) concentration 

Line 97: In this paragraph, please specify "non-halophytes" for clarity.

- The testing strategy appears somewhat unclear, given the use of both SARS-CoV-2 and HCoV-229E. It would be beneficial to explicitly mention and describe the objective within the text, such as the search for new broad-spectrum anti-coronaviral agents. 

Result

Figure 1: The scientific names of plants should be italicized. 

Line 126: “DMSO or media 126 only”. Is it “0.1% DMSO in the media” ?

Line 148-152: The scientific names of plants should be italicized. 

Line 161: provide the full name of CC50 and IC50

Line 162: delete “, CC50/IC50”

Line 170-176: remove numeric details, which presented in table 2

Table 2: Please include SI of B. halimifolia (such as >15, >89)

Figure 5: Please adjust the alignment of the figure of B. halimifolia result to match other figures

Figure 5: Please review the values of results and statistical symbols, as there appear to be inconsistencies. For example, it's unclear why the luciferase activity level (Huh-7/TMPRSS2 cells) of DCM B. halimifolia has a P-value >0.01 when it shares the same luciferase activity level with DCM S. repen (R) (P-value <0.001). Meanwhile, EtOAc S. repen (S) (Huh-7) has a higher luciferase level but a P-value <0.01. 

Line 225-230: “The ethylacetate sub-extract of Salix repens …. Huh-229 7/TMPRSS2 and Huh-7 cells, respectively.” Consider omitting this comparison since the values are similar and were not subjected to statistical comparison.

Please ensure consistency in the sub-extract names or specify them clearly to avoid confusion, such as using terms like "polar sub-extract" or " aqueous sub-extracts" or “Aq”

Line 239: remove “Similar to HCoV-229E,”

Line 307: Please specify the purity of each pure compound

Discussion

Please provide a discussion on the safety of cinnamoyl triterpenoids, considering the SI (selectivity index) values.

Materials and Methods

Please review and make necessary corrections to the methodology section for accuracy and clarity.

Line 515: Is it methylene chloride or dichloromethane?

Line 517 and 518: provide more details such as “the apolar partitions (dichloromethane and ethyl acetate parts)/ polar partition (water part)” 

4.10. Virus infection assay: Please provide clarification on the details, such as the timing of virus infection, specifically whether it occurred before, simultaneously with, or after the addition sample. Have you utilized any reference antivirus testing protocols? If so, please provide the relevant references.

Conclusions

- This section may be omitted. If it is retained, please consider using a more specific sentence to conclude the study, such as emphasizing the halophytes, the results, and the importance of the findings.

Comments on the Quality of English Language

- Please review and ensure the accurate formatting of all scientific names of plants throughout the manuscript and the supplementary. 

- Please perform a comprehensive spelling and word meaning check on the entire manuscript to ensure accuracy.

- Ensure consistency in word usage for words with similar meanings throughout the manuscript.

Author Response

Point-by-point response

This work focuses on the discovery of antiviral agents against coronaviruses from halophytes and involves the isolation and identification of their antivirals. The presented data has the potential to be valuable for the development of new antiviral drugs. However, there are some specific suggestions I would like to make to enhance the quality of the article

Introduction

Line 54- 56: “…only two drugs, paxlovid (nirmatrelvir and ritonavir) and molnupiravir, have been granted authorization by the Food and Drug Administration (FDA).” This information does not appear to be supported by the reference provided (Reference [4]). Please ensure that the source clearly supports the statement.

Answer: We thank the reviewer for this comment. We have modified the reference 4 to be in adequation with the text. (line 56).

Line 56: Please provide the full name of EMA (European Medicines Agency).

Answer: This was modified. (lines 56-57)

Line 73: Edit this to “salt (NaCl) concentration 

Answer: This was modified. (line 73)

Line 97: In this paragraph, please specify "non-halophytes" for clarity.

Answer: The sentence was slightly modified for more clarity. (99-100)

- The testing strategy appears somewhat unclear, given the use of both SARS-CoV-2 and HCoV-229E. It would be beneficial to explicitly mention and describe the objective within the text, such as the search for new broad-spectrum anti-coronaviral agents. 

Answer: We thank the reviewer for this comment. The text was modified to explicitly mention our objectives. “Hence, in our work, different halophytes and less salt-tolerant plants collected from the North of France were first screened for their antiviral activity against Human coronavirus HCoV-229E in vitro. The most active extracts and fractions were then tested against SARS-CoV-2 in order to identify potential pan-coronavirus antiviral agents. different types of coronaviruses. » (lines 91-94)

Result

Figure 1: The scientific names of plants should be italicized. 

Answer: This was modified.

Line 126: “DMSO or media 126 only”. Is it “0.1% DMSO in the media” ?

Answer: Yes, it is 0.1% DMSO in the media and it was modified. (lines 128-129)

Line 148-152: The scientific names of plants should be italicized. 

Answer: This was modified. (lines 151-155)

Line 161: provide the full name of CC50 and IC50

Answer: This was modified. (lines 171-172)

Line 162: delete “, CC50/IC50”

Answer: This was deleted and replaced by “which is the ratio between CC50 and IC50”. (lines 172-173)

Line 170-176: remove numeric details, which presented in table 2

Answer: We have simplified the numbers in the Table 2.

Table 2: Please include SI of B. halimifolia (such as >15, >89)

Answer: This was modified according to the reviewer’s suggestion.

Figure 5: Please adjust the alignment of the figure of B. halimifolia result to match other figures

Answer: the alignment was modified.

Figure 5: Please review the values of results and statistical symbols, as there appear to be inconsistencies. For example, it's unclear why the luciferase activity level (Huh-7/TMPRSS2 cells) of DCM B. halimifolia has a P-value >0.01 when it shares the same luciferase activity level with DCM S. repen (R) (P-value <0.001). Meanwhile, EtOAc S. repen (S) (Huh-7) has a higher luciferase level but a P-value <0.01. 

Answer: The values were reviewed and the statistical analysis was reconsidered. That said, we cannot compare the values between different graphs. The statistical analysis are made, graph per graph, each one having its internal control.

Line 225-230: “The ethylacetate sub-extract of Salix repens …. Huh-229 7/TMPRSS2 and Huh-7 cells, respectively.” Consider omitting this comparison since the values are similar and were not subjected to statistical comparison.

Answer: We deleted this comparison.

Please ensure consistency in the sub-extract names or specify them clearly to avoid confusion, such as using terms like "polar sub-extract" or " aqueous sub-extracts" or “Aq”

Answer: As suggested by the reviewer, we have modified the terms polar or aqueous by “Aq” in all the manuscript. We did the same with dichloromethane (DCM), ethyl acetate (EtOAc) and apolar.

Line 239: remove “Similar to HCoV-229E,”

Answer: This was removed.

Line 307: Please specify the purity of each pure compound

Answer: The purity of each compound has been specified and the chromatograms was added in the supplementary data. (line 273 and Figure S7)

Discussion

Please provide a discussion on the safety of cinnamoyl triterpenoids, considering the SI (selectivity index) values.

Answer: We thank the reviewer for this interesting remark. A paragraph was added in the discussion. (lines 474-491)

Materials and Methods

Please review and make necessary corrections to the methodology section for accuracy and clarity.

Line 515: Is it methylene chloride or dichloromethane?

Answer: It is dichloromethane, it was modified. (line 525)

Line 517 and 518: provide more details such as “the apolar partitions (dichloromethane and ethyl acetate parts)/ polar partition (water part)” 

Answer: This was modified throughout the manuscript.

4.10. Virus infection assay: Please provide clarification on the details, such as the timing of virus infection, specifically whether it occurred before, simultaneously with, or after the addition sample. Have you utilized any reference antivirus testing protocols? If so, please provide the relevant references.

Answer: Some more details were added. The products are added simultaneously with the virus. We modified the sentences (line 608 and 615). We added 2 references in which these protocols have been already used and described.

Conclusions

- This section may be omitted. If it is retained, please consider using a more specific sentence to conclude the study, such as emphasizing the halophytes, the results, and the importance of the findings.

 Answer: As suggested, we deleted the conclusion. We added a sentence at the end of the discussion (lines 494-496).

Comments on the Quality of English Language

- Please review and ensure the accurate formatting of all scientific names of plants throughout the manuscript and the supplementary. 

- Please perform a comprehensive spelling and word meaning check on the entire manuscript to ensure accuracy.

- Ensure consistency in word usage for words with similar meanings throughout the manuscript.

Answer: We have carefully checked all these different points and hope that the quality of the manuscript was improved.

Reviewer 2 Report

Comments and Suggestions for Authors

1. Manuscript title: In my opinion, it's not necessary to highlight the location where the plant materials were collected. What is so different when compared to other locations?

2. Keywords: These are key terms that did not appear in the manuscript title.

3. Figure 6: Is it possible to improve this part, because several bands are problematic.

4. Figure 8a: Significant differences between means are all missing.

5. Overall quality: Looks fine and can enrich the knowledge on exploring new drugs against SARS-CoV-2. 

Author Response

Point-by-point response

  1. Manuscript title: In my opinion, it's not necessary to highlight the location where the plant materials were collected. What is so different when compared to other locations?

Answer: Thank you for this remark. However, we prefer to mention the place of collection of the plants in the title because, depending on the place of collection, a certain phenotypical and chemical variability may exist. The plant environment and weather conditions are different depending on the location and therefore can affect the chemical composition. This can be very different from collecting a halophyte from the Mediterranean basin or the coastline of the northern France.

  1. Keywords: These are key terms that did not appear in the manuscript title.

Answer: As asked by IJMS we have “listed three to ten pertinent keywords specific to the article yet reasonably common within the subject discipline”. We think that they can be different from the title.

  1. Figure 6: Is it possible to improve this part, because several bands are problematic.

Answer: As suggested by the reviewer, we have made a new figure with another blot with clear bands.

  1. Figure 8a: Significant differences between means are all missing.

Answer: Statistical analysis have been performed and significance added on the Figure.

  1. Overall quality: Looks fine and can enrich the knowledge on exploring new drugs against SARS-CoV-2. 
